# The Role of OXT, OXTR, AVP, and AVPR1a Gene Expression in the Course of Schizophrenia

**Marta Broniarczyk-Czarniak** [1,*] **, Janusz Szemraj** [2] **, Janusz Śmigielski** [3] **and Piotr Gałecki** [1]

1   Department of Adult Psychiatry, Medical University of Lodz, 91-229 Lodz, Poland;
    piotr.galecki@umed.lodz.pl
2   Department of Medical Biochemistry, Medical University of Lodz, 92-215 Lodz, Poland;
    janusz.szemraj@umed.lodz.pl
3   Department of Health Sciences, State University of Applied Sciences in Konin, 62-510 Konin, Poland;
    janusz.smigielski.stat@gmail.com
*   Correspondence: martabroniarczyk@op.pl

**Abstract:** Schizophrenia is a serious and chronic mental illness, the symptoms of which usually appear for the first time in late adolescence or early adulthood. To date, much research has been conducted on the etiology of schizophrenia; however, it is still not fully understood. Oxytocin and vasopressin as neuromodulators that regulate social and emotional behavior are promising candidates for determining the vulnerability to schizophrenia. The aim of this study was to evaluate the expression of OXT, OXTR, AVP, and AVPR1a genes at the mRNA and protein levels in patients with schizophrenia. Due to the neurodegenerative nature of schizophrenia, the study group was divided into two subgroups, namely, G1 with a diagnosis that was made between 10 and 15 years after the onset of the illness, and G2 with a diagnosis made up to two years after the onset of the illness. Moreover, the relationship between the examined genes and the severity of schizophrenia symptoms, assessed using PANSS (Positive and Negative Syndrome Scale) and CDSS scales (Clinical Depression Scale for Schizophrenia) was evaluated. The analysis of the expression of the studied genes at the mRNA and protein levels showed statistically significant differences in the expression of all the investigated genes. OXT and AVPR1a gene expression at both the mRNA and protein levels were significantly lower in the schizophrenia group, and OXTR and AVP gene expression at both the mRNA and protein levels was higher in the schizophrenia subjects than in the controls. Furthermore, a significant correlation of OXT gene expression at the mRNA and protein levels with the severity of depressive symptoms in schizophrenia as assessed by CDSS was found.

**Keywords:** schizophrenia; oxytocin; vasopressin; OXT; OXTR; AVP; AVPR1a

## 1. Introduction

Schizophrenia is a chronic mental illness, the etiology of which is not fully understood. The lifetime prevalence of schizophrenia is estimated to be 0.5 to 1% [1]. Up to 60% of the people that are affected by schizophrenia suffer from a moderate to severe course of the illness [2]. Negative symptoms, cognitive deficits, or difficulties in social functioning, which are hard to treat, often remain after the acute psychotic symptoms have subsided [1,2].

The influence of both genetic [3] and environmental [4] factors play a significant role in the pathogenesis of schizophrenia. Such additive effects of many genes and factors [5,6] may be involved in the underlying neurodevelopmental [7–9] and neurodegenerative processes [10]. Co-occurring factors disrupt the normal developmental course in early life, leading to the manifestation of the clinical phenotype that is recognized as schizophrenia [7,11]. In the later course of the illness, more rapid brain aging processes are observed—the white matter density decreases faster than in healthy individuals, and cognitive functions deteriorate [12]. These processes are associated with the activation of inflammatory processes (IL-6) and free radicals [10].

OXT, OXTR, AVP, and AVPR1a are genes that may be involved in the development of schizophrenia [13,14]. They encode structurally similar proteins, i.e., oxytocin and vasopressin, and their receptors. These substances have a long evolutionary history of modulating a variety of stress responses—from influencing cortisol release to social interaction [15,16]. Their actions are often opposite [17]. Oxytocin facilitates prosocial behavior, increases the sense of trust and empathy [18], enhances social cognition processes and the ability to accurately distinguish facial emotions [19], promotes confidence [20–22], reduces cortisol levels, as well as behavioral reactions to stress [20–22]. Oxytocin is released centrally in response to various social and emotional stimuli and modifies human behavior [23,24]. In contrast to oxytocin, vasopressin enhances stress responses, increases anxiety, and makes faces that are depicted in pictures look more threatening [25]. Oxytocin and vasopressin acting as neuromodulators that regulate social and emotional behavior are promising candidates for determining the vulnerability to schizophrenia [13]. Since oxytocin and vasopressin appear to have physiological effects in early development in both animal and human models, it is possible that dysregulation of the oxytocin and vasopressin system causes neurodevelopmental deficits that may contribute to schizophrenia [26]. Due to its involvement in the recognition of emotions and facial expressions, oxytocin has become the starting point for research on the pathophysiology of the negative symptoms of schizophrenia, especially reduced emotional expression. Proposed hypotheses concerning the impact of oxytocin on altered social functioning include its direct coupling to dopaminergic neural pathways, hence mediating in reward and reinforcement processes in the brain, suppressing defensive behaviors, stress responses, and altering social information processing by increasing the significance of social stimuli [13,27].

The aim of this study was to evaluate the expression of OXT, OXTR, AVP, and AVPR1a genes at the mRNA and protein levels in patients who have been suffering from schizophrenia for a long or short time. Numerous studies of patients with schizophrenia indicate the possible participation of vasopressin and oxytocin in the formation of cognitive and social function deficits [28–30]. This effect may be related to the excessive activation of the hypothalamic-pituitary-adrenal (HPA) axis which is a known mechanism contributing to the occurrence of neurodegenerative changes in schizophrenia [31,32]. Due to the influence of neurodegenerative processes on the pathomechanism of schizophrenia, the study group was divided into two subgroups, i.e., G1 (10–15 years after illness onset) and G2 (up to 2 years after illness onset). The onset of schizophrenia was considered to be the moment of the first psychotic episode meeting the diagnostic criteria for schizophrenia according to DSM-5. The occurrence of the above symptoms was assessed on the basis of available medical records and an interview that was obtained from the patient or their relatives. Additionally, the relationship between the expression of the examined genes and the intensity of schizophrenia symptoms, assessed with PANSS (Positive and Negative Syndrome Scale) [33] and CDSS (Clinical Depression Scale for Schizophrenia), was investigated [34].

## 2. Materials and Methods

### 2.1. Participants

The research was conducted on a group of 106 individuals aged 18 to 65. The study group consisted of 76 people that were diagnosed with schizophrenia based on the diagnostic criteria set out in the DSM-5 classification. The study group was divided into two subgroups based on illness duration: G1—comprising of 40 patients that were suffering from schizophrenia for 10 to 15 years (mean age 38.15 ± 6.62; sex m/f 16/24); G2—comprising of 36 patients that were affected by schizophrenia for less than 2 years (mean age 29.64 ± 8.97, sex m/f 20/16). The control group (HC) consisted of 30 healthy volunteers (mean age 34.00 ± 7.91, sex m/f 11/19).

### 2.2. Safety and Exclusion Criteria

All the patients who participated in the study were hospitalized in psychiatric wards of the Specialist Psychiatric Health Care Complex in Lodz, Poland. During their hospital

stay, the patients were treated as prescribed by their treating physicians and the therapeutic process was not interfered with by their participation in the study. Individuals with intellectual disability, after severe head trauma, with coexisting serious somatic illnesses, inflammatory, and autoimmune illnesses were excluded from the study. Having read the detailed information about the study, all the subjects gave their voluntary and informed written consent to participate in the experiment. The study was approved by the Bioethics Committee of the Medical University of Lodz (No. RNN/270/18/KE).

## 2.3. Method for Assessing Schizophrenia Severity

The severity of schizophrenia symptoms was assessed once on the first day of the experiment using PANSS (Positive and Negative Syndrome Scale) [33] and CDSS (Clinical Depression Scale for Schizophrenia) [34]. The patients in the study met the criteria for the diagnosis of schizophrenia as set out in the DSM-5 classification [35].

## 2.4. Sample Collection

On the day of inclusion in the experiment, 10 mL fasting blood was collected once in the morning (7:00 a.m.) from all the subjects. The blood samples were taken by qualified medical personnel. The blood plasma was then centrifuged at 3500 rpm for 10 min; the samples were frozen at $-80\,^{\circ}$C before undergoing RNA isolation. The test samples were not thawed until biochemical analysis.

## 2.5. Identification of Serum Protein Concentration

### 2.5.1. Determination of Protein Concentration

The plasma total protein concentrations of the subjects and controls were determined with the Micro BCA™ Protein Assay Kit (Thermo Scientific, Waltham, MA, USA) according to the manufacturer's recommendations. 150 µL of reaction mixture was added to pits containing 150 µL of cytosolic fraction diluted ten-fold in 10 mM PBS buffer, pH 7.4, and then incubated (2 h, 37 $^{\circ}$C). To specify the protein concentration, the analytical curve for serum albumin was determined. Both the examined samples and the reference samples were made parallel in three repetitions. The absorbance of the samples was measured using a Multiskan Ascent microplate photometer reader (Thermo Labsystems, Philadelphia, PA, USA) at $\lambda = 570$ nm, and the total protein concentration ($\rho$) was calculated from the standard curve equation: $\rho$ protein [µg/mL] = (Abs $- 0.1048)/0.0146$. Calibration curve for albumin ($R^2 = 0.9951$).

### 2.5.2. Enzyme-Linked Immunosorbent Assay (ELISA)

The concentration of OXT, OXTR, AVP, and AVPR proteins in the serum of the subjects and the controls was determined with the following kits: Human OXT Elisa kit MyBioSource (San Diego, CA, USA), Human OXTR Elisa kit MyBioSource (San Diego, CA, USA), Human AVP Elsa Kit MyBioSource (San Diego, CA, USA), and Human AVPR1a Elisa Kit Antibodies (Aachen, Germany), according to the protocols that were provided by the manufacturer. β-Actin was used as an endogenous control for protein concentration in the samples and assayed with the Human Actin Beta (ACTb) ELISA Kit (BMASSAY, Singapore) according to the manufacturer's recommendations. A total of 100 µL of cytosolic fraction ($\rho$ protein = 0.5 mg/mL) was added to pits that were coated with antibodies specific for the proteins analyzed, and then incubated (1.5 h, 37 $^{\circ}$C). The contents were removed, and the pits were washed in three shifts in 10 mM PBS buffer, and then incubated (1 h, 37 $^{\circ}$C) with 100 µL of biotinylated antibodies specific for the proteins analyzed. The contents were then removed, and the pits were washed in three shifts in 10 mM PBS buffer, and then incubated (30 min, 37 $^{\circ}$C) with 100 µL ABC Working Solution. The contents were removed, and the pits were washed in five shifts in 10 mM PBS buffer, and then incubated (10 min, 37 $^{\circ}$C) with 90 µL of TMB substrate. After adding 100 µL of TMB Stop Solution, the absorbance of the samples was measured using a Multiskan Ascent microplate photometer reader

(Thermo Labsystems) at $\lambda = 450$ nm. To specify the protein concentration, analytical curves for the analyzed proteins were prepared.

### 2.6. Evaluation of Expression of Selected Genes at mRNA Level

#### 2.6.1. Total RNA Isolation

Isolation of total RNA from the patients' and control subjects' blood was performed with the InviTrap Spin Universal RNA Kit (Stratec Molecular, Berlin, Germany) according to the manufacturer's instructions. A total of 300 μL of blood in a tube was incubated with 300 μL of Lysis/Binding Buffer; 300 μL of acid phenol:chloroform mixture was added to the cell lysate, and, after mixing, the sample was centrifuged (5 min, $10,000 \times g$) to separate the aqueous phase from the organic phase. The top (aqueous) fraction was transferred to a fresh tube containing 375 μL of 96% ethanol and, after mixing, the mixture was transferred to a tube with a filter column. After centrifugation (15 s, $10,000 \times g$), the filter column was moved to fresh tubes and washed in 700 μL of RNA Wash Solution 1, and then centrifuged (10 s, $10,000 \times g$). The filter column was washed twice in 500 μL of Wash Solution 2/3 and centrifuged (1 min, $10,000 \times g$). The filter column was placed in a fresh tube and the isolated RNA was subject to elution in 30 μL of water free from nucleases with temperature of 95 °C by means of centrifugation (30 s, $10,000 \times g$). The absorbance was measured using a spectrophotometer (Picodrop) at $\lambda = 260$ nm and total RNA concentration was read from the table. The isolated RNA was stored at −80 °C.

#### 2.6.2. Isolated RNA Quality Analysis

The quality of total RNA was checked with the Agilent RNA 6000 Nano Kit (Agilent Technologies, Santa Clara, CA, USA) in accordance with the manufacturer's recommendations. A total of 1 μL RNA 6000 Nano dye was added to a tube containing 65 μL Agilent RNA 6000 Nano gel matrix, and centrifuged (10 min, $13,000 \times g$). The mixture of gel and fluorescent dye was applied on the surface of a Nano chip that was placed in the workstation. Then, 5 μL of RNA Nano marker was added to selected pits. The isolated RNA samples and the RNA size marker were denatured (2 min, 70 °C) and then 1 μL of the sample was pipetted into the indicated pits of the Nano chip and mixed (1 min, 2400 rpm). The quality of the isolated RNA was checked using the 2100 Bioanalyzer (Agilent Technologies). The level of total RNA degradation was determined with the use of an electrophoretogram and the obtained RIN values (the RNA integrity number). Only the samples with an RIN value > 7 were subject to further analysis.

#### 2.6.3. RT-PCR Reverse Transcription

RT reaction was performed with TaqMan® RNA Reverse Transcription Kit (Applied Biosystems, Waltham, MA, USA) according to the manufacturer's instructions using specific starters and probes Hs 00792417g1, Hs 00168573m1, Hs 00356994g1, and Hs 00176122m1 for OXT, OXTR, AVP, and AVPR1a gene, respectively, provided by Applied Biosystems. The samples were incubated (30 min at 16 °C and 30 min at 42 °C) in a thermocycler (Biometra, Jena, Germany). Reverse transcriptase was inactivated (5 min at 85 °C) and the obtained cDNA was stored at −20 °C.

#### 2.6.4. Real-Time PCR Reaction

Real-Time PCR reaction was conducted using the TaqMan® Universal PCR Master Mix, No UNG (Applied Biosystems) according to the protocol that was provided by the manufacturer. To calculate the relative gene expression at the mRNA level, the Ct Livak 2001 comparative method was used. The expression levels of OXT, OXTR, AVP, and AVPR1a genes in the individual samples were normalized against the reference gene RPL13A. To calculate the relative gene expression, the comparative CT method was used. The CT values were read from the amplification curve. The levels of gene expression in particular samples were normalized against the reference gene RPL13A.

### 2.6.5. Statistical Analysis

Statistical and graphical analyses were performed using Statistica 13 software. The lowest level of statistical significance totaled $p < 0.05$. Statistical analysis of the tested non-measurable variables was performed based on the non-parametric Pearson's chi-squared test. Statistical analysis of the measurable variables was performed on the basis of non-parametric tests since the analysis of empirical distributions of the studied parameters based on the Shapiro–Wilk test did not demonstrate compliance with normal distribution. Levene's test was used to assess the homogeneity of variance. Using an analysis of variance in addition to conformance to the normal distribution is possible provided there are no grounds for rejecting the hypothesis of equality of variances. The non-parametric Mann–Whitney $U$ test was used in this study to compare the investigated groups. In addition, the Kruskal–Wallis non-parametric test and Dunn's test were used. Due to the distributions of the examined characteristics and their nature, Spearman's rank correlation coefficient was applied in this study.

### 3. Results

The statistical analysis showed no significant difference ($p > 0.05$) in terms of gender according to the study group. The G1 and G2 study groups differed in age ($p < 0.001$), but neither the G1 nor the G2 group was statistically significantly different in age from the control group (HC).

The severity of schizophrenia symptoms as assessed by PANSS and CDSS did not differ significantly between the study groups (PANSS—Positive Scale Z = −0.07, $p = 0.938$; PANSS—Negative Scale Z = 1.11, $p = 0.266$; PANSS—General Scale Z = 0.47, $p = 0.636$; PANSS Z = 0.81, $p = 0.420$; CDSS Z = 0.15, $p = 0.880$).

### 3.1. OXT Gene Expression at mRNA and Protein Levels

The mRNA expression level of the OXT gene was statistically significantly different between all study groups (H = 71.96; $p < 0.001$). Dunn's multiple comparison test showed that the mRNA expression level was statistically significantly lower in G1 ($p < 0.001$) and G2 ($p < 0.001$) than in the control group (HC) and was statistically significantly lower in G1 than in G2 groups ($p = 0.008$) (Table 1, Figure 1).

**Table 1.** OXT, OXTR, AVP, and AVPR1a gene expression at the mRNA level. * Statistically significant difference between all groups; ** statistically significant difference between the control group; *** statistically significant difference between the second study group; $p$—statistical significance; median (minimum–maximum), ±—standard deviation.

| | G1 | G2 | HC | *p*-Value |
|---|---|---|---|---|
| OXT $2^{\Delta ct}$ | 0.17 (0.12–0.23) ± 0.03 * | 0.19 (0.14–0.29) ± 0.03 * | 0.32 (0.25–0.38) ± 0.04 | G1:HC—$p < 0.001$; G2:HC—$p < 0.001$; G1:G2 $p = 0.008$ |
| OXTR $2^{\Delta ct}$ | 0.49(0.40–0.56) ± 0.04 * | 0.42 (0.37–0.50) ± 0.03 * | 0.34 (0.17–0.39) ± 0.05 | G1:HC—$p < 0.001$; G2:HC—$p < 0.001$; G1:G2—$p = 0.000$ |
| AVP $2^{\Delta ct}$ | 0.96 (0.71–1.15) ± 0.11 ** | 0.90 (0.83–0.99) ± 0.05 ** | 0.80 (0.62–0.99) ± 0.09 | G1:HC—$p < 0.001$; G2:HC—$p < 0.001$; G1:G2—$p > 0.05$ |
| AVPR1a $2^{\Delta ct}$ | 0.72 (0.67–0.79) ± 0.03 * | 0.74 (0.70–0.80) ± 0.03 *** | 0.79 (0.65–0.86) ± 0.06 | G1:HC—$p < 0.001$; G2:HC—$p > 0.05$; G1:G2—$p = 0.001$ |

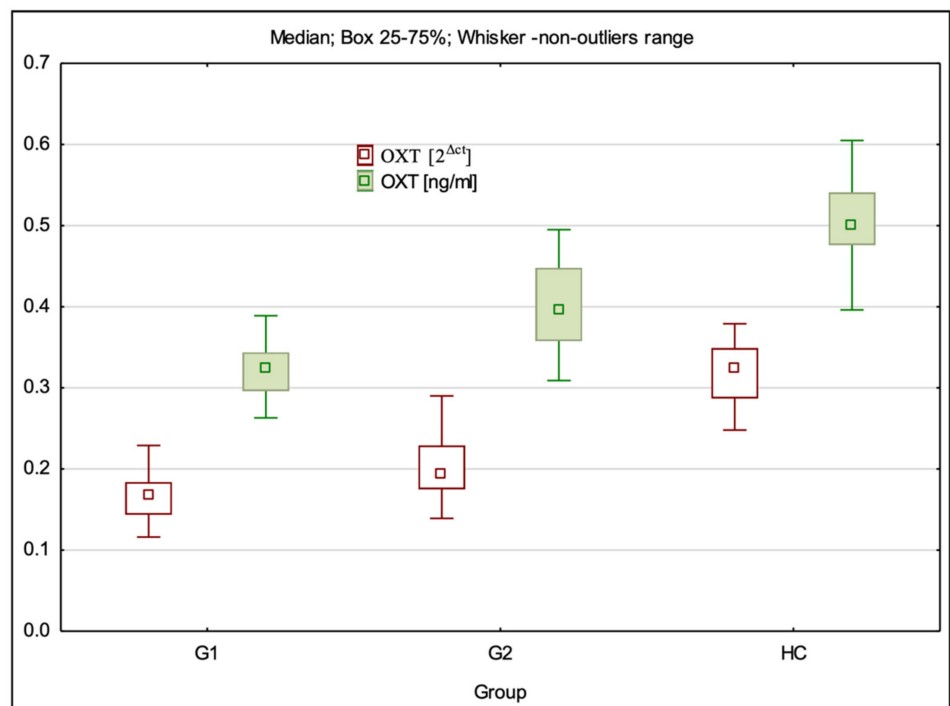

**Figure 1.** OXT gene expression at the mRNA and protein levels in the studied population.

The level of OXT expression at the protein level was statistically significantly different between all study groups (H = 78.05; $p < 0.001$). Dunn's multiple comparison test showed that the OXT protein level was statistically significantly lower in G1 ($p < 0.001$) and G2 ($p < 0.001$) than in the control group and was significantly lower in G1 than in G2 groups ($p < 0.001$) (Table 2, Figure 1).

**Table 2.** OXT, OXTR, AVP, and AVPR1a gene expression at the protein level. * Statistically significant difference between all groups; ** statistically significant difference between the control group; *** statistically significant difference between the second study group; $p$—statistical significance; median (minimum–maximum), ±—standard deviation.

|  | G1 | G2 | HC | *p*-Value |
|---|---|---|---|---|
| OXT ng/mL | 0.32 (0.26–0.39) ± 0.03 * | 0.40 (0.31–0.50) ± 0.05 * | 0.50 (0.40–0.61) ± 0.05 | G1:HC—$p < 0.001$; G2:HC—$p < 0.001$; G1:G2 $p < 0.001$ |
| OXTR ng/mL | 0.32 (0.24–0.57) ± 0.07 * | 0.26 (0.21–0.33) ± 0.03 * | 0.15 (0.10–0.26) ± 0.05 | G1:HC—$p < 0.001$; G2:HC—$p < 0.001$; G1:G2 $p < 0.001$ |
| AVP ng/mL | 0.89 (0.54–0.97) ± 0.11 ** | 0.74 (0.61–0.84) ± 0.05 ** | 0.63 (0.46–0.82) ± 0.09 | G1:HC—$p < 0.001$; G2:HC—$p < 0.001$; G1:G2 $p > 0.05$ |
| AVPR1a pg/mL | 0.55 (0.51–0.75) ± 0.04 * | 0.58 (0.54–0.64) ± 0.03 *** | 0.64 (0.49–0.70) ± 0.03 | G1:HC—$p < 0.001$; G2:HC—$p > 0.05$; G1:G2 $p = 0.004$ |

### 3.2. OXTR Gene Expression at mRNA and Protein Levels

The mRNA expression level of OXTR was statistically significantly different between all the study groups (H = 78.61; $p < 0.001$). Dunn's multiple comparison test showed that the OXTR level was statistically significantly higher in G1 ($p < 0.001$) and G2 ($p < 0.001$)

than in the control group and was significantly higher in G1 than in G2 groups ($p < 0.001$) (Table 1, Figure 2).

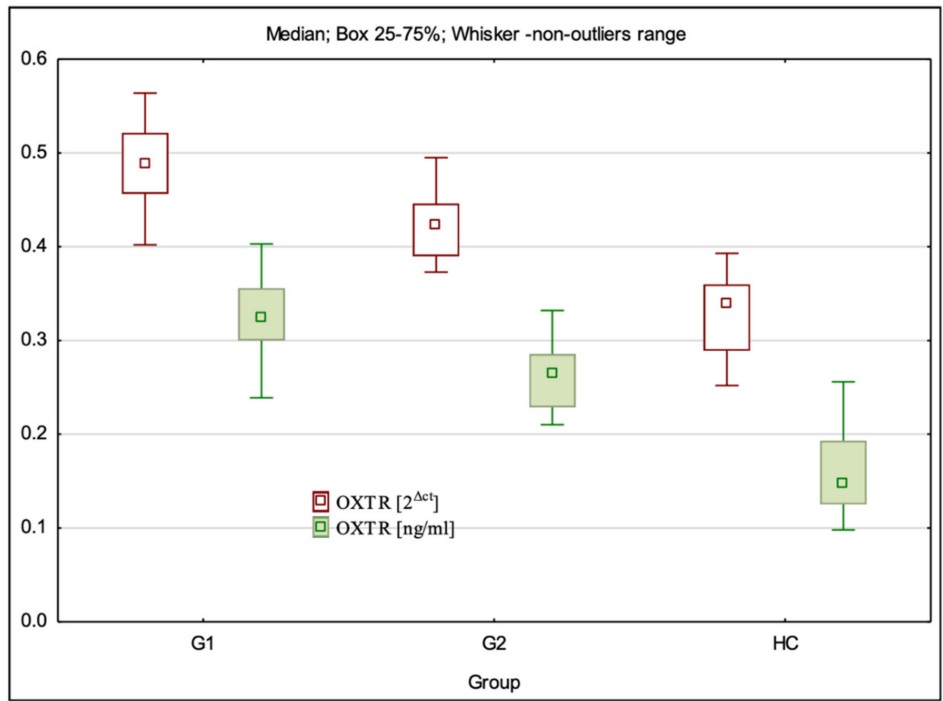

**Figure 2.** OXTR gene expression at the mRNA and protein levels in the studied population.

The level of OXTR expression at the protein level was statistically significantly different between all the study groups (H = 74.86; $p < 0.001$). Dunn's multiple comparison test showed that the OXTR protein level was statistically significantly higher in G1 ($p < 0.001$) and G2 ($p = 0.000$) than in the control group (HC) and was significantly higher in G1 than in G2 groups ($p = 0.000$) (Table 2, Figure 2).

*3.3. AVP Gene Expression at the mRNA and Protein Levels*

The mRNA expression level of AVP was statistically significantly different between the examined groups (H = 38.08; $p < 0.001$). Dunn's multiple comparison test showed that the mRNA expression level was statistically significantly higher in G1 ($p < 0.001$) and G2 ($p < 0.001$) than in the control group. The mRNA expression of AVP was not statistically significantly different between the study groups G1 and G2 ($p > 0.05$) (Table 1, Figure 3).

The level of AVP expression at the protein level was statistically significantly different between the investigated groups (H = 38.00; $p < 0.001$). Dunn's multiple comparison test showed that the AVP levels were statistically significantly higher in G1 ($p < 0.001$) and G2 ($p < 0.001$) than in the control group (HC). The AVP protein levels were not different between G1 and G2 groups ($p > 0.05$) (Table 2, Figure 3).

*3.4. AVPR1a Gene Expression at mRNA and Protein Levels*

The mRNA expression level of AVPR1a was statistically significantly different between the examined groups (H = 32.03; $p < 0.001$). Dunn's multiple comparison test showed that the mRNA expression level of AVPR1a was statistically significantly lower in G1 ($p < 0.001$) than in the control group (HC). AVPR1a mRNA expression in G1 was statistically significantly lower than in G2 ($p = 0.001$). There was no statistically significant difference in AVPR1a mRNA expression between G2 and HC groups ($p > 0.05$) (Table 1, Figure 4).

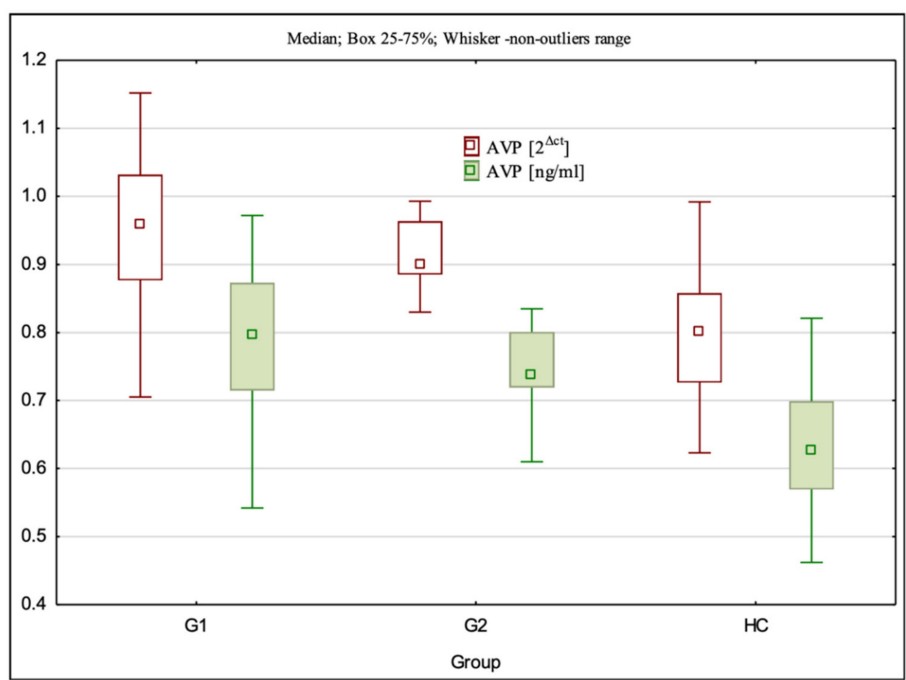

**Figure 3.** AVP gene expression at the mRNA and protein levels in the studied population.

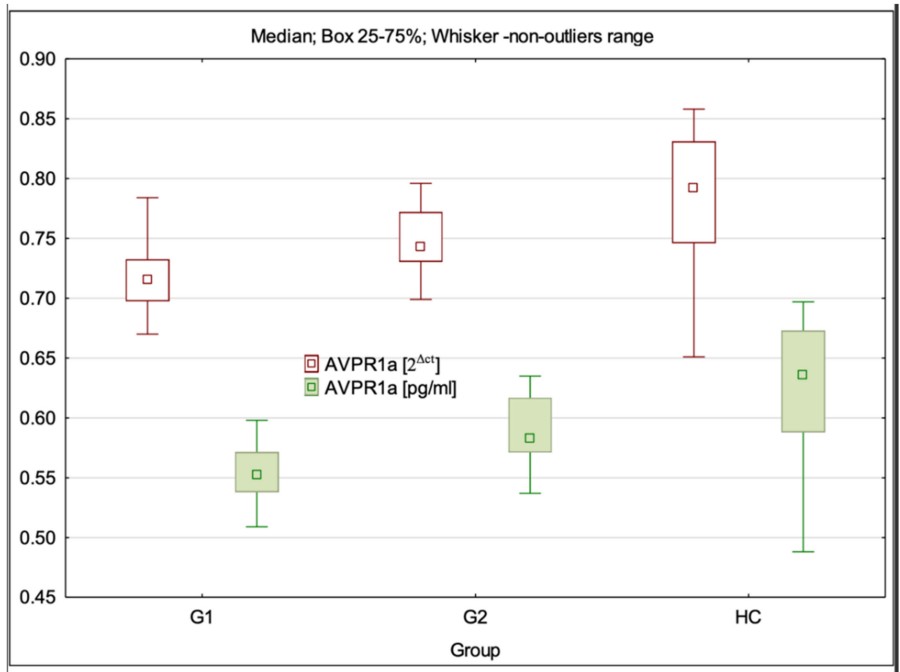

**Figure 4.** AVPR1a gene expression at the mRNA and protein levels in the studied population.

The level of AVPR1a expression at the protein level was statistically significantly different between the investigated groups (H = 28.36; $p < 0.001$). Dunn's multiple comparison test showed that the AVPR1a levels were statistically significantly lower in G1 ($p < 0.001$) than in the control group (HC). AVPR1a protein level in G1 was statistically significantly lower than in G2 ($p = 0.004$). There was no statistically significant difference in AVPR1a expression at the protein level between G2 and HC groups ($p > 0.05$) (Table 2, Figure 4).

### 3.5. Analysis of Correlation of OXT, OXTR, AVP, and AVPR1a Gene Expression at mRNA and Protein Levels with the Severity of Schizophrenia Symptoms Assessed by PANSS and CDSS

Spearman's rank correlation coefficient analysis in G2 showed a statistically significant correlation of OXT gene expression at the mRNA (OXT $2^{\Delta ct}$) and protein (OXT ng/mL) levels with the severity of depressive symptoms as assessed by CDSS (q = 0.42; $p$ = 0.011 and q = 0.37; $p$ = 0.028, respectively). In the remaining cases, no statistically significant ($p$ > 0.05) correlation of the values of expression of the examined genes with the severity of the individual symptoms of schizophrenia was found.

## 4. Discussion

In recent years, many studies have been conducted to assess the differences in the levels of oxytocin and vasopressin in patients with schizophrenia. Most of the published studies have assessed the level of plasma oxytocin as an indicator of its function in the central nervous system. This procedure may raise doubts as to the validity of such a measurement and its implications. According to a meta-analysis by Valstad et al., despite the blood-brain barrier, there is a correlation between the central and peripheral oxytocin levels [36]. The statistical analyses comparing the expression values of OXT, OXTR, AVP, and AVPR1a genes in the investigated groups allow us to conclude that all the examined genes may participate in the etiopathogenesis of schizophrenia. The results indicate decreased activity of the oxytocin system and increased activity of the vasopressin system in this group of patients. The OXT gene expression values at the mRNA level and OXT protein levels decrease, while the OXTR levels increase with illness duration. The neuromodulatory effect of oxytocin may, therefore, be an important pathogenetic factor in schizophrenia [29,37,38]. Considering the human central nervous system, OXTR receptor expression is observed in the corpus striatum, black matter, the amygdala, and the hippocampus. These areas of the brain are associated with the pathophysiology of schizophrenia [39]. Increasing the availability of ligand for the receptor for a prolonged period of time can cause a down-regulation of the receptor [40,41]. There is limited evidence that such a relationship between the concentration of oxytocin in the extracellular fluid in specific brain regions or in the plasma and the expression of OXTR exists—the results of this study suggest that it does occur. Similar studies, limited to patients with a first episode of schizophrenia, were conducted by Liu Yong et al. [42] and Yang et al. [43]. Their results indicate the presence of lower plasma concentrations of OXT and OXTR proteins and higher expression of OXT and OXTR at the mRNA level in the patient group. The results regarding the expression of OXT at the mRNA level and the concentration of plasma OXTR protein [42] are contrary to the results that were obtained. The differences may suggest the occurrence in the early symptomatic stage of the illness (when experiencing the first psychotic episode) of a compensatory mechanism, the aim of which is to increase the oxytocin production by enhancing OXT gene expression at the mRNA level. This mechanism may become 'exhausted' shortly after the development of the illness, which may translate into a decrease in OXT expression at the mRNA level, a drop in plasma oxytocin levels, and an increase in the frequency of negative symptoms and cognitive deficits, which rises with the duration of psychosis [29,30]. Admittedly, Spearman's rank analysis showed no association of OXT and OXTR gene expression with the severity of schizophrenia symptoms that were assessed using PANSS, which may indicate that the genes that were studied have no effect on the symptoms revealed. On the other hand, studies that were conducted in other centers indicate that such relationships do exist. An association of lower oxytocin levels with difficulties in the correct identification of facial emotions, as well as with developmental defects of the hippocampus and amygdala [34,44], with an increased number of stressful life events, with higher intensity of negative symptoms [29], with less severe positive symptoms [45–47], with more severe cognitive deficits [30], and higher levels of anxiety in patients with schizophrenia is demonstrated most frequently [48]. Studies have also confirmed the therapeutic effects of intranasally administered oxytocin on symptoms of schizophrenia [27,49,50].

The research results revealed a relationship between the OXT gene expression at the mRNA and protein levels with the severity of depressive symptoms that were assessed by CDSS in the group of patients who have been suffering from schizophrenia for a short time. The importance of oxytocin in the development of depressive symptoms has been the subject of many studies. Oxytocin seems to play a role especially in the development of postpartum depression symptoms [51,52]. Research suggests a potential effect of exogenous oxytocin administered to patients during labor on exacerbating the course of postpartum mood disorders [51]. Oxytocin is considered an antistress and antidepressant substance; on the other hand, it may contribute to the dysregulation of the hypothalamic-pituitary-adrenal (HPA) stress axis [53,54]. Studies on oxytocin levels in patients suffering from depression are inconclusive. Some indicate an association of lower oxytocin levels with greater severity of depressive symptoms in MDD [54,55], others conversely confirm the presence of higher oxytocin levels in depressed patients [56] and the correlation of higher serum oxytocin levels with an elevated risk of anxiety symptoms, greater sensitivity to stress, and a greater risk of depression in pregnant women [56,57]. No effect of antidepressants on oxytocin levels has been demonstrated [55]. Antipsychotic drugs, on the other hand, do not affect or increase oxytocin release [58].

Research results indicate that there is an increased activation of the vasopressin system in people with schizophrenia. The AVP gene expression values at the mRNA level and AVP protein plasma levels are higher, while AVPR1a is lower in schizophrenic patients than in the control subjects. The lack of a significant difference in AVP gene expression between long- and short-term patients may suggest that the involvement of vasopressin in the etiopathogenesis of schizophrenia is a phenomenon that is occurring from the beginning of the illness and does not change with time. Vasopressin stimulates and regulates the HPA axis and subsequently leads to increased cortisol secretion [32]. Overactivation of the vasopressin system may cause an overactivation of the HPA stress axis, which is a known mechanism that contributes to neurodegenerative changes in schizophrenia [31]. Due to its multiple occurrences in the brain, the AVPR1a receptor plays an important role in the impact of vasopressin on behavior, memory and social cognition, pair bonding, parental care, sexual behavior, and aggressive behavior [59,60]. In addition, it has been shown to be involved in the synthesis and release of cortisol, regulation of circadian rhythm, pain, and adaptive behavior [61–63]. Alterations in this area may play a role in the emergence and increase of cognitive and social function deficits that are associated with HPA axis overactivation [28].

The findings on AVPR1a mRNA gene expression are consistent with the study that was conducted by Yang et al. [43]—they showed no significant difference in gene expression in the people that were suffering from schizophrenia for a short time and the control group. The difference in the AVPR1a gene expression was statistically significant only in long-term patients. These results allow a conclusion to be drawn that changes in the expression of the major cerebral vasopressin receptor, i.e., AVPR1a, may play a role in the pathogenesis of schizophrenia, especially in the development of cognitive deficits that also increase with the duration of the illness [28]. The research results that were collected by Yang et al., indicate that the AVP gene mRNA expression is lower in individuals with a first episode of schizophrenia [43], which is contrary to the results of this study.

The research showed no association of AVP and AVPR1a gene expression with the severity of schizophrenia symptoms. The available literature indicates that such a relationship, namely higher vasopressin levels in people with schizophrenia, correlated with greater severity of positive symptoms [64], cognitive deficits [28], and with an increased response of the HPA axis to stress [39,44]. Contradictory information is provided by data which indicate that vasopressin administered intranasally reduced the severity of schizophrenia symptoms that were assessed using PANSS, including cognitive functions—mainly related to long- and short-term memory [65]. In another study, intranasally administered desmopressin—a synthetic analogue of vasopressin—reduced the severity of negative symptoms in patients with schizophrenia [66,67].

In summary, the results indicate that the oxytocin-vasopressin system is involved in the pathogenetic processes accompanying schizophrenia [29,68]. Available data indicate the involvement of neurohormones in the manifestation of specific groups of illness symptoms and the effects on stress axis activity [27,29,31]. There is no evidence that would allow clear conclusions to be drawn, but some indicate an important role of the studied genes in the pathogenesis of the illness. Assessing the activity of the oxytocin-vasopressin system is an interesting target for further studies on the pathophysiology of schizophrenia. The results that were obtained should be interpreted with caution given the limitations of the study. The patients in the study were taking antipsychotic drugs, hence their potential influence on the study results cannot be excluded. In a study on rats, Uvnäs-Moberg et al., indicate that some neuroleptics can increase oxytocin release (clozapine, amperozide) or have no effect (haloperidol, raclopride) on oxytocin release—the study involved both peripheral blood and cerebrospinal fluid analyses [58]. The study groups are not very large, so the study should be repeated on a larger cohort of patients. The stratification factor should also be considered, and studies should be extended to other patient populations.

## 5. Conclusions

The expression of OXT, OXTR, AVP, and AVPR1a genes at the mRNA and protein levels differ between the studied groups of patients with schizophrenia and the control group. OXT and AVPR1a gene expression was significantly lower in the schizophrenia group than in the controls, whereas OXTR and AVP gene was significantly higher in the schizophrenia subjects than in the controls. Moreover, the expression of OXT, OXTR, and AVPR1a genes both at the mRNA and protein levels differed statistically significantly in the group of individuals that were suffering from schizophrenia for a long and short period. The expression of the AVP gene doesn't differ between the long-term and short-term patients. OXT gene expression at the mRNA and protein levels additionally correlated positively with the severity of depressive symptoms as assessed by CDSS. The results indicate the potential involvement of the genes that were studied in both the neurodevelopmental and neurodegenerative processes underlying schizophrenia.

**Author Contributions:** Conceptualization, M.B.-C. and P.G.; Methodology, M.B.-C. and J.S. (biochemical methodology).; Formal analysis, M.B.-C., P.G., J.Ś. (statistical analysis); Investigation, M.B.-C. and J.S. (biochemical analysis); Writing—original draft preparation, M.B.-C.; Writing—review and editing, M.B.-C., P.G.; Visualization, J.Ś.; Funding acquisition, P.G. All authors have read and agreed to the published version of the manuscript.

**Funding:** This work was supported by the Medical University of Lodz, Poland [Research Program No. 503/5-062-02/503-51-001-19-00]. The funders had no role in the study design, data and literature collection and analysis, decision to publish, or the preparation of the manuscript.

**Institutional Review Board Statement:** The study was approved by the Bioethics Committee of the Medical University of Lodz (No. RNN/270/18/KE).

**Informed Consent Statement:** Having read the detailed information about the study, all the subjects gave their voluntary and informed written consent to participate in the experiment.

**Data Availability Statement:** The data analyzed in the study are available upon request to the authors of the article.

**Conflicts of Interest:** The authors declare no conflict of interest.

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
