# Peer review of "The Role of OXT, OXTR, AVP, and AVPR1a Gene Expression in the Course of Schizophrenia"

_cimb, doi:10.3390/cimb44010025_

Round 1

Reviewer 1 Report

This manuscript focuses on a study on the role of OXT, OXTR, AVP, AVPR1a gene expression in the course of schizophrenia. Although the manuscript has a good quality, there are different issues that lead to discourage the publication in its current form.

ABSTRACT:

Generally well written, with a few minor grammatical flows. Lines 24 and 25 “protein levels WAS” -> “Protein Levels WERE”. I also suggest specifying the acronyms PANSS and CDSS.

INTRODUCTION:

Generally well written.

Lines 38-41:

“Both neurodevelopmental [3-5] and neurodegenerative processes [6] contribute to the onset of schizophrenia. Genetic [7] and environmental [8] factors have an impact on their development. The additional action of multiple genes and factors plays a key role in the pathogenesis of the illness [9, 10].”

-> The style is too monotonous and repetitive. Consider rewriting it.

Study hypothesis lines 70-77:

“The aim of this study was to evaluate the expression of OXT, OXTR, AVP, AVPR1a genes at mRNA and protein levels in patients who have been suffering from schizophrenia for a long or short time. Due to the influence of neurodegenerative processes on the pathomechanism of schizophrenia, the study group was divided into two subgroups, i.e.,  G1 (10-15 years after illness onset) and G2 (up to 2 years after illness onset). Additionally, the relationship between the expression of the examined genes and the intensity of schizophrenia symptoms, assessed with PANSS (Positive and Negative Syndrome Scale) [28] and CDSS (Clinical Depression Scale for Schizophrenia), was investigated [29].”

-> This section should be explained more in detail. Why should the expressions of these genes vary over time? How were the thresholds decided?

MATERIALS AND METHODS:

Lines 166-168:

“The level of total RNA degradation was determined with the use of an electrophoretogram (figure) and the obtained RIN values. Only the samples with RIN value > 7 were subject to further analysis.”

-> The Acronym “RIN” must be explained as well as the rationale behind the choice of the >7 threshold.

Line 171:

Manufacturer’ -> Manufacturer’s

Line 192:

To assess homogeneity of variance à to assess THE homogeneity of variance

RESULTS:

No major issue was found.

Tables and main text: p values = 0.000 should be < 0.001

DISCUSSION:

Lines 279-282:

“In recent years, there have been many studies evaluating differences in oxytocin and vasopressin levels in patients with schizophrenia. Most published studies have assessed plasma oxytocin levels as an indicator of its function in the central nervous system. Such a procedure may raise questions about the validity of measurement and its implications.” -> These sentences are unclear, hard to follow and with a few major grammar mistakes.

CONCLUSIONS:

This section should be discussed more in detail.

GRAPHS AND FIGURES:

All good.

REFERENCES:

All good.

REVIEWER’S CONCLUSION:

Minor revision needed. Check also the punctuation of the entire manuscript.

Author Response

Dear Reviewer,

thank you for your comments on the manuscript. I tried to respond to all your comments.  Changes are included in the attached file.

Kind regards

Marta Broniarczyk-Czarniak

Reviewer 2 Report

In the present manuscript entitled, "The role of OXT, OXTR, AVP, AVPR1a gene expression in the course of schizophrenia” , Broniarczyk-Czarniak et al. explored the role of OXT, OXTR, AVP, AVPR1a in schizophrenia through the gene expression analyses such as Real time PCR and ELISA experiments. The major conclusions put forth by the authors are that OXT  and AVPR1a gene expression was significantly reduced in patients. I think that the manuscript is excellently written. Therefore, it raises concerns whether these results are significant and represent an important contribution to the psychiatric genetics field.

Author Response

Dear Reviewer

I am very grateful for the positive evaluation of the manuscript. 

Kind Regards

Marta Broniarczyk-Czarniak

Round 2

Reviewer 1 Report

I suggest approving the article in its current form.